# Predictive Radiographic Values for Foot Ulceration in Persons with Charcot Foot Divided by Lateral or Medial Midfoot Deformity

**DOI:** 10.3390/jcm11030474

**Published:** 2022-01-18

**Authors:** Mateo López-Moral, Raúl J. Molines-Barroso, Irene Sanz-Corbalán, Aroa Tardáguila-García, Marta García-Madrid, José Luis Lázaro-Martínez

**Affiliations:** Diabetic Foot Unit, Facultad de Medicina, Instituto de Investigación Sanitaria del Hospital Clínico San Carlos (IdISSC), Universidad Complutense de Madrid, 28040 Madrid, Spain; matlopez@ucm.es (M.L.-M.); irsanz01@ucm.es (I.S.-C.); aroa_tg@hotmail.com (A.T.-G.); magarc28@ucm.es (M.G.-M.); diabetes@ucm.es (J.L.L.-M.)

**Keywords:** diabetic foot, Charcot neuroarthropathy, midfoot deformity, radiographic measures

## Abstract

Background: To identify differences in radiographic outcomes in weight-bearing lateral X-ray to predict the probability of ulceration in patients with midfoot Charcot neuroarthropathy (CN) differentiated by lateral and medial column deformities. Methods: Thirty-five patients who suffered from CN midfoot deformity participated in this 1 year prospective study in a specialized diabetic foot unit. Lateral talar-first metatarsal angle, calcaneal pitch, and cuboid height were performed by digital radiographs in the weight-bearing lateral view. Patients were followed up for 1 year or until an ulcer ulceration event occurred in the midfoot region. Results: ROC analyses showed that all patients with medial pattern deformity that developed a midfoot ulcer had a lateral talar-first metatarsal angle greater (negative) than −27.5 degrees (°). All patients with lateral pattern deformity that developed a midfoot ulcer had a calcaneal pitch greater (more negative) than −5° and a cuboid height greater (more negative) than −1.5°. Conclusions: Lateral talar-first metatarsal angle was the greatest predictor of midfoot ulceration, with greater than −27.5° measurement correlating with ulceration occurrence in patients with medial deformity. Calcaneal pitch and cuboid height were the greatest predictors of midfoot ulceration with greater than −5 and −1.5°, respectively in patients with CN lateral deformity.

## 1. Introduction

Charcot neuroarthropathy (CN) is a progressive, noninfectious, destructive inflammatory process of the foot and ankle, leading to severe deformities and further risk of amputation and death [1,2], with diabetes and polyneuropathy being the leading causes [3,4,5].

According to the anatomical classification [5], midfoot deformity is the most common location, accounting for up to 60% of patients with CN diagnosis [6,7,8].

Midfoot Charcot patients are characterized by bony prominences in the medial or lateral aspect of the foot, increasing plantar pressures and predisposing them to ulcer occurrence in the midfoot [9]. Patients with Charcot-related foot ulcers have 12-times greater probability of foot amputation than those without foot ulcers [1,10].

The pathophysiology of CN has been previously described as medial arch collapse (medial pattern involvement) followed by further lateral arch collapse (lateral pattern involvement) [11]. In the affected Charcot foot, authors have suggested that patients with medial and lateral CN patterns are presented differently [12]. Sagittal deformities are more prone to develop an ulcer occurrence than transverse plane deformities [11]. Pinzur et al. [13] found that patients with varus CN pattern who underwent midfoot reconstruction surgery were least likely to achieve a favorable clinical outcome.

One of the most important goals in CN management is to prevent foot ulceration and further deep infections such as osteomyelitis; to that end, clinicians should consider performing radiographic analyses to know the severity of the affected feet. Additionally, the radiographic assessment in correlation with the clinical severity of CN deformity is useful to assist surgeons in deciding when to operate [14]. Previous authors [15,16] have found that the most reproducible angles to assess CN severity are talar first metatarsal, calcaneal pitch, and cuboid height. A previous systematic review [12] has shown that a direct relation exists between radiographic measurements and ulcer occurrence; despite this, results are heterogeneous between different authors as authors did not separately analyze varus and valgus deformities [11,17].

Nonetheless, no previous studies have prospectively evaluated whether lateral or medial column pattern deformity could affect the prognosis of patients with CN with midfoot affection.

Therefore, this study aimed to identify differences in radiographic outcomes in a weight-bearing lateral X-ray to predict the probability of ulceration in patients with midfoot Charcot foot differentiated by lateral and medial column deformities.

## 2. Materials and Methods

### 2.1. Subjects

Thirty-five patients at risk for foot ulceration and who suffered from CN midfoot deformity participated in this 1 year prospective study in a specialized diabetic foot unit between December 2018 and January 2021.

The inclusion criteria were confirmed type 1 or type 2 diabetes, age > 18 years, affected with chronic CN stage 3 according to Eichenholtz classification (defined as consolidation of deformity, joint arthrosis, fibrous ankyloses, rounding and smoothing of bone fragments) [18] affecting the midfoot location, and loss of protective foot sensation because of peripheral neuropathy. All the patients were included at least 4 weeks after CN stage 3 was diagnosed and the inflammatory process was stable.

Exclusion criteria ulcers during the examination, history of rheumatoid disease, other causes of neuropathy, critical limb ischemia as defined according to the International Working Group Diabetic Foot (IWGDF) guidance [19], and the need for walking aids. Additionally, patients with reconstructive or offloading surgery in the midfoot were also excluded.

After institutional review, board approval was obtained, and the medical records and clinicopathologic conditions of patients were analyzed.

### 2.2. Clinical Evaluation

At baseline, clinical characteristics were assessed after the patient signed informed consent on day 0. Midfoot CN deformity was defined as a bony midfoot prominence of the foot [20]; the apex of the deformity is located at the midtarsal joint and was identified radiographically by signs of previous bone fracture or luxation, according to Eichenholtz classification [19]. CN midfoot deformity pattern was stratified into a medial column pattern (if the clinical weight-bearing pattern of the heel was in valgus and the forefoot was normal or abducted, with medial arch convexity), and a lateral column pattern (if the heel was clinically aligned in varus with weight bearing, they walked on the outer border of the involved foot, and the forefoot was adducted relative to the hindfoot, with lateral arch convexity) as described by Pinzur [13].

Clinicopathologic data were collected, including diabetes type, mean duration of diabetes and CN disease, hypertension, and HbA1c (%) values over the previous 3 months. BMI was calculated as weight (kg) divided by height (m^2^).

Patients’ renal, cardiac, and retinopathy statuses and previous minor amputations were recorded in the case report. Systolic toe and ankle pressures were registered. Additionally, ankle brachial index (ABI) and toe brachial index (TBI) values were calculated for the Charcot limb.

According to the IWGDF guidelines, critical limb ischemia was defined as the absence of both distal pulses and a brachial ankle index of <0.39, systolic ankle pressure < 50 mmHg, and toe pressure < 30 mmHg [19].

Diabetic polyneuropathy (DPN) was diagnosed according to the inability to sense the pressure of a 10 g Semmes–Weinstein monofilament at three plantar foot sites and/or a vibration perception threshold > 25 V as assessed using a biothesiometer (Me.Te.Da. s.r.l., Via Silvio Pellico, 4, 63074 San Benedetto del Tronto, Italy) [21].

### 2.3. Radiographic Measurement

The three most reliable radiographic measures to assess for foot ulceration were performed by digital radiographs in the lateral view, including the lateral talar-first metatarsal angle, calcaneal pitch, and cuboid height [11,12]. All radiographic measurements were taken from the weight-bearing position at baseline [16].

The lateral talar-first metatarsal angle was measured as the angle formed from a line bisecting the talar body and neck and a line bisecting the first metatarsal. Apex plantar angulation was considered a negative angle [16]. The calcaneal pitch was measured as the angle between the reference line from the plantar aspect of the calcaneal tuberosity to the plantar aspect of the fifth metatarsal head and a line extending from the most plantar aspect of the calcaneal tuberosity to the most plantar aspect of the anterior process of the calcaneus. Cuboid height was measured as the perpendicular distance from the plantar aspect of the cuboid to a line drawn from the plantar aspect of the calcaneal tuberosity to the plantar aspect of the fifth metatarsal head. The distance was measured in millimeters and was negative if the plantar cuboid was plantar to this line [16] (Figure 1). Two trained clinicians with more than 5 years of experience treating diabetic foot complications analyzed and extracted data from each radiographic measurement and calculated the average angle. Both investigators who analyzed and extracted data from radiographic angles were blinded to the clinical data from every patient.

### 2.4. Follow-Up

All patients were followed up for 1 year or until they suffered from an ulcer ulceration event in the midfoot region. Patients came monthly to the outpatient clinic, according to the international guidelines [22]. In every visit, the principal investigator performed debridement of high-risk points, such as minor lesions, defined as nonulcerative lesions of the skin on the plantar aspect of the foot and included abundant callus, hemorrhage, or a blister [23]. Additionally, all the patients wore an extra-depth custom-made shoe with a total contact insole to decrease peak pressures in the plantar aspect of the foot.

### 2.5. Outcome Measures

The main outcome measure was to select the optimal cut-off point on the scale of radiographic measurement (talar-first metatarsal angle, calcaneal pitch angle, and cuboid height) that has an optimum combination of sensitivity and specificity to predict for neuropathic ulceration in the midfoot of patients with CN separated by lateral or medial pattern.

Midfoot ulceration was defined according to the IWGDF guidelines as a break of the skin of the foot that involves, as a minimum, the epidermis and part of the dermis [24]. The investigator who assessed DFU ulceration was blinded to the radiographic measurement to avoid bias in goniometric interpretation.

The secondary outcome measure assessed differences in the midfoot ulceration risk between lateral or medial CN pattern patients. Midfoot minor lesion was defined as a foot lesion that has a high risk of developing into a foot ulcer, such as intracutaneous or subcutaneous hemorrhage, blister, or skin fissure not penetrating into the dermis in a person at risk [24].

### 2.6. Statistical Analyses

The assumption of normality of all continuous variables was verified using the Kolmogorov–Smirnov test. Normally distributed variables (Kolmogorov–Smirnov test with *p* ≥ 0.05) were reported as mean and standard deviations (SD), and non-normally distributed variables (Kolmogorov–Smirnov test with *p* < 0.05) were reported as medians and interquartile ranges. The Chi-square test for categorical variables and the Student T-test for quantitative variables were performed to explore differences in clinical features between patients with and without midfoot ulcer occurrence. To select the optimal diagnostic cut-off points on the scale of radiographic analyses, ROC curves were used. This is a graphical method of representing sensitivity and specificity for a given test. In addition, for those radiographic measurements with sensitivity and specificity less than 100%, positive predictive value (PPV), negative predictive value (NPV), positive likelihood ratio (PLR), and negative likelihood ratio (NLR) were calculated for lateral talar-first metatarsal angle, calcaneal pitch angle, and cuboid height. *p*-values < 0.05 were considered statistically significant, with confidence intervals (CI) of 95%. All statistical analyses were performed using SPSS statistics version 25.0 for Mac OS (SPSS, Chicago, IL, USA).

## 3. Results

Thirty-five patients with chronic CN midfoot deformity were included. All the patients included presented no ulcers at the time of inclusion in the study. Patients were followed up prospectively for a 1 year period or until they developed an ulceration event in the midfoot. Baseline data on demographic characteristics and diabetes complications are shown in Table 1.

From the thirty-five included patients, 19 (54.3%) patients presented lateral column pattern deformity, and 16 (45.7%) patients presented medial column pattern deformity.

Regarding baseline characteristics and CN duration before inclusion, we did not find any difference between groups. Radiographic measurements were statistically different between both lateral and medial pattern groups (Table 1).

During the follow-up period, 16 (45.7%) patients suffered from a midfoot ulceration event in a median duration time of 4 [Interquartile Range (25–75th); 3–7.5] weeks. Patients with lateral column pattern suffered from more midfoot minor lesions (*n* = 15, 75%) compared to medial column pattern patients (*n* = 6, 40%) (*p* = 0.036); additionally, patients with lateral column pattern were more prone (*n* = 12, 60%) to develop a midfoot ulceration event in comparison with midfoot ulceration patients (*n* = 4, 26.7%).

### Main Outcome

Lateral talar-first metatarsal angle showed an association with ulcer occurrence in both patterns, medial column pattern (*p* < 0.001 CI (5.73–11.54)), and lateral column pattern (*p* = 0.015 CI (0.85–6.89)). Calcaneal pitch angle was associated with midfoot ulceration in those patients with a lateral column pattern affection *p* < 0.001 CI (5.03–8.21)). Cuboid height was associated with ulceration in those patients with a lateral column pattern (*p* < 0.001 CI (3.13–5.28)) (Table 2).

Using ROC analyses, we found that all patients with medial pattern deformity that developed a midfoot ulcer had a lateral talar-first metatarsal angle greater (negative) than −27.5 degrees (Sensitivity = 100; Specificity = 100%). All patients with lateral pattern deformity that developed a midfoot ulcer had a calcaneal pitch greater (more negative) than −5 degrees and a cuboid height greater (more negative) than −1.5 degrees (Sensitivity = 100%; Specificity = 100%) (Table 3).

## 4. Discussion

The results derived from the current research demonstrated that radiographic measurements that predicted midfoot ulceration in patients with CN midfoot deformity varied depending on the lateral or medial presentation of Charcot’s foot.

Values of −5° for calcaneal pitch angle and −1.5 mm for cuboid height predicted 100% of the cases of midfoot ulceration in patients with a lateral pattern, and values of −27.5° for talar-first predicted 100% of the cases of midfoot ulceration in patients with a medial pattern. Additionally, patients with lateral column patterns showed a higher risk of midfoot ulceration and minor lesions than medial column pattern patients.

The data from this research support the assessment of calcaneal pitch angle and cuboid height in a weight-bearing radiographic measurement for lateral column pattern patients and the assessment of talar-first metatarsal angle for medial column pattern patients due to the optimum levels of sensitivity and specificity to predict DFU on the midfoot of patients with CN midfoot deformity.

In parallel to our results, Bevan and Wukich et al. [11,17] compared CN patients with and without ulcerations to determine if their radiographic measurements were related to diabetic foot ulceration. Bevan et al. found that patients with ulceration had a talar-first metatarsal angle below −27°, similar to those obtained for Wukich et al. (−32.9°). These results are higher than we found in the current research; we found that 100% of CN patients with a lateral talar-first metatarsal angle below −27.5° developed midfoot ulceration in the group of patients with medial column pattern deformity. The difference of 5° with previous research could be explained because they did not separately analyze lateral and medial column patterns. For calcaneal pitch angle, Bevan et al. found that patients with foot ulceration had −11° and Wukich et al. found that patients with foot ulceration had 6.3°. Our results are similar to those found in Bevan et al.’s research [17] because we found that calcaneal pitch angle values below −5° were a perfect predictor of midfoot ulceration for lateral column pattern patients. In the current investigation, cuboid height was a lateral column pattern ulceration predictor for radiographic values below −1.5 mm. Bevan et al. showed that patients with DFU had 9 mm of lateral column height, and Wukich found that patients with DFU had −5.1 mm of cuboid height. The results found in the current research are different compared with previous research [11,17] because they did not separately analyze lateral and medial column pattern deformities.

We found that 60% of patients with lateral column pattern deformity ulcerated compared to only 26.7% of medial column pattern. Previous research [14] found that lateral column ulcers are much more difficult to treat, so the clinical relevance of separately analyzing lateral and medial column patterns remains very important. Pinzur et al. [13] demonstrated that lateral column patients who underwent reconstructive foot surgery had worse outcomes than medial column pattern patients.

Surgeons and clinicians should consider separately analyzing radiographic values for medial and lateral column pattern patients; when radiographic affection is present, clinicians should consider these patients as nonplantigrade foot and unstable due to their high risk for foot ulceration. Reconstructive foot surgeries must be implemented in patients with a history of DFU and radiographic alterations in the weight-bearing position.

To our knowledge, this is the first study to investigate radiographic cut-off values to predict neuropathic ulceration in patients with CN midfoot deformity analyzed separately for the clinical presentation by lateral or medial pattern.

However, our results should be interpreted with caution because of some limitations; the current study only analyzed the three most-studied radiographic measurements [12]; further research should confirm the benefit of other radiographic analyses for preventive and operative therapies. Additionally, we only had a small sample size, although past studies had commonly small sample sizes due to the low prevalence of CN disease [14,16,17]. Additionally, previous research [25] has found that plain X-rays are not reliable when differentiating CN from osteomyelitis; further research should confirm it by using leukocyte imaging. Despite our patients did not show any statistical difference between groups regarding glycated hemoglobin, this biomarker has previously shown to play a central role in the diagnosis and follow-up of patients with diabetes [26]. Further research should confirm this fact, differentiated by different Charcot patterns and foot ulceration.

Since the first wave of the COVID-19 pandemic was very overt in Spain from February 2020, all patients included in this study were free of COVID-19 infection, since it is well known that COVID-19 seems to be a paramount contributor for diabetic foot lesions due to increased cytokine levels [27].

## 5. Conclusions

Lateral talar-first metatarsal angle is the greatest angular prediction of midfoot ulceration, with greater than −27.5° measurement correlating with ulceration occurrence in patients with medial pattern deformity. At the same time, calcaneal pitch and cuboid height are the greatest angular predictors of midfoot ulceration, greater than −5° and −1.5°, respectively in patients with Charcot lateral pattern deformity.

## Figures and Tables

**Figure 1 jcm-11-00474-f001:**
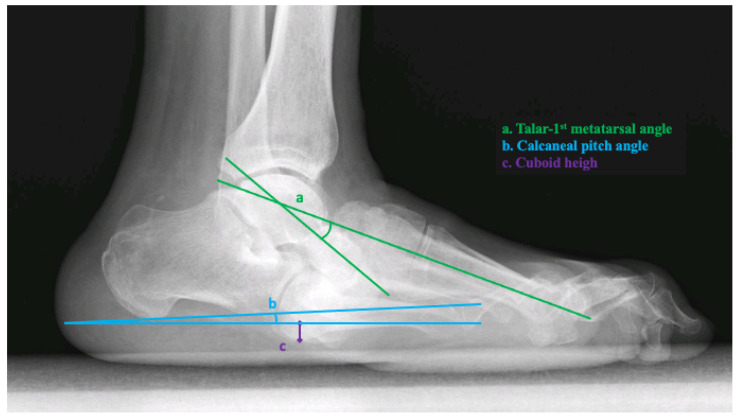
Radiographic measurements in the weight-bearing lateral X-ray. Legend. a, talar-first metatarsal angle (°); b, calcaneal pitch angle (°); c, cuboid height (mm).

**Table 1 jcm-11-00474-t001:** Differences between the risk factors for lateral and medial Charcot midfoot patterns.

Baseline Characteristics	Patients (*n* = 35)	Lateral Column Pattern Patients (*n* = 19)	Medial Column Pattern Patients (*n* = 16)	*p*-Value
Men, *n* (%)	32 (91.4%)	17 (89.4%)	16 (100%)	0.117
Women *n*, (%)	3 (8.6%)	2 (11.6%)	-	
Type 2 Diabetes, *n* (%)	33 (94.3%)	19 (95%)	14 (93.3%)	0.833
Retinopathy, *n* (%)	27 (77.1%)	14 (70%)	13 (86.7%)	0.245
Nephropathy, *n* (%)	15 (42.9%)	8 (40%)	7 (46.7%)	0.693
Hypertension, *n* (%)	27 (77.1%)	15 (70%)	13 (86.7%)	0.245
Hypercholesterolemia, *n* (%)	30 (85.7%)	16 (80%)	14 (93.3%)	0.265
Previous minor Amputation, *n* (%)	22 (62.9%)	10 (50%)	12 (80%)	0.069
Ankle Brachial Pressure Index, mean ± SD	1.23 ± 0.32	1.27 ± 0.26	1.17 ± 0.39	0.426
Toe Brachial Pressure Index, mean ± SD	0.86 ± 0.28	0.96 ± 0.26	0.77 ± 0.29	0.397
Mean age ± SD (years)	62.54 ± 10.36	60.05 ± 9.62	65.87 ± 10.69	0.101
Body mass index (kg/cm^2^), mean ± SD	28.74 ± 5.32	30.12 ± 3.71	26.89 ± 6.6	0.076
Glycated hemoglobin mmol/mol (%), mean ± SD	7.53 ± 1.34	7.64 ± 1.39	7.38 ± 1.29	0.571
Diabetes mellitus (years), mean ± SD	17.23 ± 8.88	16.35 ± 10.87	18.4 ± 5.35	0.508
Duration of Charcot foot prior to inclusion (years), mean ± SD	6.37 ± 4.59	5.4 ± 3.53	7.66 ± 5.58	0.152
Calcaneal pitch angle (°), mean ± SD	−4.45 ± 5.5	−6.1 ± 3.74	−2.26 ± 6.75	0.004 *
Cuboid height (mm), mean ± SD	−0.31 ± 2.91	−2.15 ± 2.39	2.13 ± 1.3	<0.001 *
Talar-1st metatarsal angle (°), mean ± SD	−20.54 ± 5.48	−17.45 ± 3.69	−24.66 ± 4.77	<0.001 *

Legend: DM, diabetes mellitus; SD, standard deviation; Kg, kilograms; cm^2^, squared centimeters; mm, millimeters; °, degrees. * *p* < 0.05 indicates statistical significance. Statistical significance refers to differences between CN patterns.

**Table 2 jcm-11-00474-t002:** Differences between radiographic measurements for CN patterns and the presence of midfoot ulceration.

Charcot Pattern	Radiographic Measurement	Midfoot Ulceration	Non-Midfoot Ulceration	*p*-Value
Lateral column pattern (*n* = 19)	Calcaneal pitch angle (°), mean ± SD	−8.75 ± 1.91	−2.12 ± 1.45	<0.001 *
	Cuboid height (mm), mean ± SD	−3.83 ± 1.2	0.37 ± 1.06	<0.001 *
	Talar-first metatarsal angle (°), mean ± SD	−19 ± 3.38	−15.12 ± 2.94	0.015 *
Medial column pattern (*n* = 16)	Calcaneal pitch angle (°), mean ± SD	−8.5 ± 11.03	0 ± 2.48	0.221
	Cuboid height (mm), mean ± SD	2 ± 0.81	2.18 ± 1.47	0.181
	Talar-first metatarsal angle (°), mean ± SD	−31 ± 1.82	−22.36 ± 3.01	<0.001 *

Legend: SD, standard deviation; mm, millimeters; °, degrees. * *p* < 0.05 indicates statistical significance.

**Table 3 jcm-11-00474-t003:** Performance characteristics of radiographic measurements for lateral and medial Charcot midfoot patterns.

Charcot Pattern	Pooled	Calcaneal Pitch Angle	Cuboid Height	Talar-1st Metatarsal Angle
Lateral column pattern (*n* = 19)	Cut-off point	−5°	−1.5 mm	−17.5°
	AUC	1	1	0.797
	Sensitivity	100	100	75
	Specificity	100	100	87.5
	PPV	-	-	0.9
	NPV	-	-	0.7
	PLR	-	-	6
	NLR	-	-	0.28
Medial column pattern (*n* = 16)	Cut-off point	−1.5°	1.5 mm	−27.5°
	AUC	0.68	0.51	1
	Sensitivity	50	75	100
	Specificity	45	63.6	100
	PPV	0.24	0.42	-
	NPV	1.62	1.69	-
	PLR	0.9	2	-
	NLR	1.1	0.4	-

Legend: AUC, area under curve; PPV, positive predictive value; NPV, negative predictive value; PLR, positive likelihood ratio; NLR, negative likelihood ratio; °, degrees; mm, millimeters.

## Data Availability

The data are available via request to the corresponding author.

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
