# Peer review of "Predictive Radiographic Values for Foot Ulceration in Persons with Charcot Foot Divided by Lateral or Medial Midfoot Deformity"

_jcm, 2022, doi:10.3390/jcm11030474_

Round 1

Reviewer 1 Report

Dear authors,
The study tries to show the radiographic predictive values of foot ulceration in people with Charcot foot divided by lateral or medial deformity of the midfoot. Considering an important issue being a health priority.
I point out several questions:
1) In the abstract it is necessary to suppress spaces in the text.
2) There are up-to-date scientific articles that should appear in the theoretical framework.
3) In the material and methods section, I consider it important to detail and explain the inclusion criteria.
4) Line 150 is missing a point at 0.05.

Rest without incident. Congratulations,

Author Response

Respond to reviewers

Reviewer 1

Dear reviewer thank you very much for your comments and corrections, we have tried to respond all the issues. We know that all your appreciations will improve the quality of the paper. See in bold all the respond to your comments. You can see in yellow highlight all the changes in the revised manuscript.

The study tries to show the radiographic predictive values of foot ulceration in people with Charcot foot divided by lateral or medial deformity of the midfoot. Considering an important issue being a health priority.
I point out several questions:

1) In the abstract it is necessary to suppress spaces in the text.

Thank you for your comment, we have checked the abstract and suppressed all the spaces in the text accordingly.

2) There are up-to-date scientific articles that should appear in the theoretical framework.

Thank you very much for your comment, this issue was checked, we have added the following scientific articles, lines 32 and 35.

  • Harkin EA, Schneider AM, Murphy M, Schiff AP, Pinzur MS. Deformity and Clinical Outcomes Following Operative Correction of Charcot Ankle. Foot Ankle Int. 2019;40:145-151.
  • Rettedal D, Parker A, Popchak A, Burns PR. Prognostic Scoring System for Patients Undergoing Reconstructive Foot and Ankle Surgery for Charcot Neuroarthropathy: The Charcot Reconstruction Preoperative Prognostic Score. J Foot Ankle Surg. 2017; 57:17-28.

3) In the material and methods section, I consider it important to detail and explain the inclusion criteria.

Thank you very much for the comment, we have added the explanation of CN stage 3 according to Eichenholtz classification: “The inclusion criteria were confirmed type 1 or type 2 diabetes, age > 18 years, affected with chronic CN stage 3 according to Eichenholtz classification (defined as consolidation of deformity, joint arthrosis, fibrous ankyloses, rounding and smoothing of bone fragments) [18] affecting the midfoot location, and loss of protective foot sensation because of peripheral neuropathy. All the patients were included at least 4 weeks after CN stage 3 was diagnosed and the inflammatory process was stable.” Lines 70 to 74.

4) Line 150 is missing a point at 0.05.

Typo error was corrected accordingly.

Rest without incident. Congratulations,

Reviewer 2 Report

Dear Authors,

It was a pleasure to read and review the manuscript “Predictive radiographic values for foot ulceration in persons 2 with Charcot foot divided by lateral or medial midfoot deformity”.

Authors clearly highlight the fact that Lateral talar-1st metatarsal angle is correlated with ulceration occurrence in patients with medial pattern deformity. At the same time, calcaneal pitch and cuboid 260 height are the greatest angular predictors of midfoot in patients with Charcot lateral pattern deformity as they mention in their conclusion. However, they should reconstruct their manuscript to highlight the gold standard of imaging diagnostic procedure. More specific:

A crucial point is the stage that the diagnostic procedure by imaging (E.g MR imaging, plain radiography) was performed since the observational study by Chantelau et al mentions “MR images revealed traumatic bone and joint injuries (bone oedema, occult fractures, and joint effusion) already in stage 0, when X-ray still showed normal bone and joint anatomy (p = 0.02). Moreover, MR images revealed bone oedema, joint effusion and soft tissue oedema in addition to fractures and calluses in stage I (bone dissolution), stage II (bone coalescence), and stage III (bone remodeling), i.e., in stages with overt radiographic pathology”. (Chantelau E, Poll LW. Evaluation of the diabetic charcot foot by MR imaging or plain radiography--an observational study. Exp Clin Endocrinol Diabetes. 2006 Sep;114(8):428-31.)

You should clarify the interval that the diagnostic procedure was performed, since radiographic appearance may appear negative for a few days up to 3 weeks. In particular, is it the same for all 35 subjects of the study?

Another point is that often in Charcot neuropathy, osteomyelitis is present. In these cases “Plain X-rays are normal in acute COA as opposed to typical changes seen in gouty and rheumatoid arthritis. Nevertheless, plain X-rays are not reliable in

order to differentiate COA from osteomyelitis. Instead, nuclear modalities (phase technetium-99m methylene diphosphonate and indium-111-labeled leukocyte scintigraphy) may be required.” This is also mentioned in recent bibliography (Lauri C, Glaudemans AWJM, Signore A. Leukocyte Imaging of the Diabetic Foot. Curr Pharm Des. 2018;24(12):1270-1276.)

You should discuss the levels of HbA1c alone and in correlation with Hematocrit. More specific, HbA1c levels may impact and provoke diabetic complications even in early stages (Kaiafa G. et al. Is HbA1c an ideal biomarker of well-controlled diabetes? Postgrad Med J. 2021 Jun;97(1148):380-383). Please mention in table 1 the mean levels of HBA1c and possible RBC and plasma transfusions. Did the patients receive transfusion of RBCs or plasma as supportive therapy, since it is well known that it is associated with the healing process of ulcers.

Since the first wave of COVID-19 Pandemic was very overt in Spain from February 2020, authors should mention if all 35 patients included in this study were free of COVID-19 infection, since it is well known that COVID-19 seems to be a paramount contributor for diabetic foot lesions, due to increased cytokine levels (International Journal of Lower Extremity Wounds. 2020;19(2): 111)

Author Response

Reviewer 2

Dear reviewer thank you very much for your comments and corrections, we have tried to respond all the issues. We know that all your appreciations will improve the quality of the paper. See in bold all the respond to your comments. You can see in yellow highlight all the changes in the revised manuscript.

It was a pleasure to read and review the manuscript “Predictive radiographic values for foot ulceration in persons 2 with Charcot foot divided by lateral or medial midfoot deformity”.

Authors clearly highlight the fact that Lateral talar-1st metatarsal angle is correlated with ulceration occurrence in patients with medial pattern deformity. At the same time, calcaneal pitch and cuboid 260 height are the greatest angular predictors of midfoot in patients with Charcot lateral pattern deformity as they mention in their conclusion. However, they should reconstruct their manuscript to highlight the gold standard of imaging diagnostic procedure. More specific:

A crucial point is the stage that the diagnostic procedure by imaging (E.g MR imaging, plain radiography) was performed since the observational study by Chantelau et al mentions “MR images revealed traumatic bone and joint injuries (bone oedema, occult fractures, and joint effusion) already in stage 0, when X-ray still showed normal bone and joint anatomy (p = 0.02). Moreover, MR images revealed bone oedema, joint effusion and soft tissue oedema in addition to fractures and calluses in stage I (bone dissolution), stage II (bone coalescence), and stage III (bone remodeling), i.e., in stages with overt radiographic pathology”. (Chantelau E, Poll LW. Evaluation of the diabetic charcot foot by MR imaging or plain radiography--an observational study. Exp Clin Endocrinol Diabetes. 2006 Sep;114(8):428-31.)

You should clarify the interval that the diagnostic procedure was performed, since radiographic appearance may appear negative for a few days up to 3 weeks. In particular, is it the same for all 35 subjects of the study?

Dear reviewer, thank you very much for the comment, the protocol of our diabetic foot clinic is to prescribe a therapeutic footwear and custom-made insole, when consolidation of the CN process is established, it means absence of inflammatory signs and symptoms and at least 4 weeks of CN stage 3 diagnosis. We take monthly X-ray to monitor the coalescent stage. We have added the interval that the diagnostic procedure was performed: “All the patients were included at least 4 weeks after CN stage 3 was diagnosed and the inflammatory process was stable.” Lines 70 to 74.

Another point is that often in Charcot neuropathy, osteomyelitis is present. In these cases “Plain X-rays are normal in acute COA as opposed to typical changes seen in gouty and rheumatoid arthritis. Nevertheless, plain X-rays are not reliable in order to differentiate COA from osteomyelitis. Instead, nuclear modalities (phase technetium-99m methylene diphosphonate and indium-111-labeled leukocyte scintigraphy) may be required.” This is also mentioned in recent bibliography (Lauri C, Glaudemans AWJM, Signore A. Leukocyte Imaging of the Diabetic Foot. Curr Pharm Des. 2018;24(12):1270-1276.)

Thank you very much for the comment, we have discussed it as a limitation: “Additionally, previous research [25] have found that plain X-rays are not reliable in order to differentiate CN from osteomyelitis, further research could confirm it by using leukocyte imaging.” Lines 279 to 281.

You should discuss the levels of HbA1c alone and in correlation with Hematocrit. More specific, HbA1c levels may impact and provoke diabetic complications even in early stages (Kaiafa G. et al. Is HbA1c an ideal biomarker of well-controlled diabetes? Postgrad Med J. 2021 Jun;97(1148):380-383). Please mention in table 1 the mean levels of HBA1c and possible RBC and plasma transfusions. Did the patients receive transfusion of RBCs or plasma as supportive therapy, since it is well known that it is associated with the healing process of ulcers.

Thank you for your comment, we have discussed the role of HbA1c in the healing prognosis and it relationship with CN: “Despite our patients did not show any statistical difference between groups regarding glycated Haemoglobyn, this biomarker has previously showed to be a central role in the diagnosis and follow-up of patients with diabetes [25], further research should confirm this fact differentiated by different charcot patterns and foot ulceration.” Lines 281 to 285.

Glycated Haemoglobyn did not show any difference between groups:

Baseline Characteristics

Patients (N=35)

Lateral column pattern patients

(n=19)

Medial column pattern patients

(n=16)

P-value

Glycated hemoglobin mmol/mol (%), mean ± SD

7.53 ± 1.34

7.64 ± 1.39

7.38 ± 1.29

0.571

Hematocrit was not analyzed in the study. Additionally, the patients included the current study did not receive transfusion of RBCs or plasma as supportive therapy, we will consider it for further research and clinical practice.

Since the first wave of COVID-19 Pandemic was very overt in Spain from February 2020, authors should mention if all 35 patients included in this study were free of COVID-19 infection, since it is well known that COVID-19 seems to be a paramount contributor for diabetic foot lesions, due to increased cytokine levels (International Journal of Lower Extremity Wounds. 2020;19(2): 111)

Thank you very much for the comment, it will improve the quality of the paper. We have discussed it accordingly: “Since the first wave of COVID-19 Pandemic was very overt in Spain from February 2020, all patients included in this study were free of COVID-19 infection, since it is well known that COVID-19 seems to be a paramount contributor for diabetic foot lesions, due to increased cytokine levels [27].” Lines 286 to 289.

Round 2

Reviewer 2 Report

Authors have responded to all my comments!

I am happy with the current version.